# Transcriptional Profiling of Malignant Melanoma Reveals Novel and Potentially Targetable Gene Fusions

**DOI:** 10.3390/cancers14061505

**Published:** 2022-03-15

**Authors:** Sourat Darabi, Andrew Elliott, David R. Braxton, Jia Zeng, Kurt Hodges, Kelsey Poorman, Jeff Swensen, Basavaraja U. Shanthappa, James P. Hinton, Geoffrey T. Gibney, Justin Moser, Thuy Phung, Michael B. Atkins, Gino K. In, Wolfgang M. Korn, Burton L. Eisenberg, Michael J. Demeure

**Affiliations:** 1Hoag Family Cancer Institute, Newport Beach, CA 92663, USA; david.braxton@hoag.org (D.R.B.); burton.eisenberg@hoag.org (B.L.E.); michael.demeure@hoag.org (M.J.D.); 2Caris Life Sciences, Phoenix, AZ 85040, USA; aelliott@carisls.com (A.E.); jzeng@carisls.com (J.Z.); khodges@carisls.com (K.H.); kpoorman@carisls.com (K.P.); jswensen@carisls.com (J.S.); usbasava@gmail.com (B.U.S.); jhinton@carisls.com (J.P.H.); wmkorn@carisls.com (W.M.K.); 3Lombardi Comprehensive Cancer Center, MedStar Georgetown University Hospital, Washington, DC 20007, USA; geoffrey.t.gibney@gunet.georgetown.edu (G.T.G.); mba41@georgetown.edu (M.B.A.); 4Honor Health Research Institute, Scottsdale, AZ 85258, USA; jmoser@honorhealth.com; 5Department of Pathology, University of South Alabama, Mobile, AL 36617, USA; tphung@health.southalabama.edu; 6Division of Oncology, Norris Comprehensive Cancer Center, University of Southern California, Los Angeles, CA 90033, USA; gino.in@med.usc.edu; 7Translational Genomics Research Institution, Phoenix, AZ 85004, USA

**Keywords:** melanoma, tumor molecular profiling, biomarkers, next-generation sequencing, whole transcriptome sequencing, oncogenic fusions

## Abstract

**Simple Summary:**

Malignant melanoma is a complex disease that is estimated to claim over 7000 lives in the United States in 2021. Although recent advances in genomic technology have helped with the identification of driver variants, molecular studies and clinical trials have often focused on prevalent alterations, such as the *BRAF*-V600E mutation. With the inclusion of whole transcriptome sequencing, molecular profiling of melanomas has identified gene fusions and revealed gene expression profiles that are consistent with the activation of signaling pathways by common driver mutations. Patients harboring such fusions may benefit from currently approved targeted therapies and should be considered in the design of future clinical trials to further personalize treatments for patients with malignant melanoma.

**Abstract:**

Invasive melanoma is the deadliest type of skin cancer, with 101,110 expected cases to be diagnosed in 2021. Recurrent *BRAF* and *NRAS* mutations are well documented in melanoma. Biologic implications of gene fusions and the efficacy of therapeutically targeting them remains unknown. Retrospective review of patient samples that underwent next-generation sequencing of the exons of 592 cancer-relevant genes and whole transcriptome sequencing for the detection of gene fusion events and gene expression profiling. Expression of PDL1 and ERK1/2 was assessed by immunohistochemistry (IHC). There were 33 (2.6%) cases with oncogenic fusions (14 novel), involving *BRAF*, *RAF1*, *PRKCA*, *TERT*, *AXL*, and *FGFR3*. MAPK pathway-associated genes were over-expressed in *BRAF* and *RAF1* fusion-positive tumors in absence of other driver alterations. Increased expression in tumors with *PRKCA* and *TERT* fusions was concurrent with MAPK pathway alterations. For a subset of samples with available tissue, increased phosphorylation of ERK1/2 was observed in *BRAF*, *RAF1*, and *PRKCA* fusion-positive tumors. Oncogenic gene fusions are associated with transcriptional activation of the MAPK pathway, suggesting they could be therapeutic targets with available inhibitors. Additional analyses to fully characterize the oncogenic effects of these fusions may support biomarker driven clinical trials.

## 1. Introduction

Malignant melanoma is a clinically and biologically complex disease; while recent advances have led to the identification of genomic driver alterations that can be targeted with small molecule therapeutics, some melanomas do not harbor targetable genomic alterations or will display either primary or acquired resistance to targeted therapy. Further research toward the development of novel therapeutic strategies is needed. Gene fusions result from a genomic rearrangement that joins the sequences of independent genes, which often leads to abnormal expression and function of the resulting fusion proteins. Genomic rearrangements that lead to gene fusions can take place via several mechanisms, including inversions, translocations, duplications, and deletions [1]. Oncogenic gene fusions have been identified in several hematological malignancies and solid tumors, including melanomas, and the impact of these fusions is variable and not fully understood [2]. Fusions have proven to be attractive drug targets in the case of *ALK*, *NTRK*, and *FGFR* fusions, leading to the development and approval of efficacious novel drugs for other cancers [3].

The cancer genome atlas (TCGA) PanCancer Atlas reported over 400 patients with cutaneous melanomas, with mutations most commonly observed in *BRAF*. A total of 145 samples harbored fusions, and the most common fusions were seen in *RAF1* and *BRAF* genes [4,5]. Furthermore, over 300 fusions were reported in AACR Project GENIE among a melanoma cohort of 3800 samples, both from primary and metastatic lesions, including in-frame or out-of-frame fusions. From primary lesions, only 89 fusions were identified. The remainder of the fusions detected were from the 217 metastatic site samples, of which 41 harbored *BRAF* fusions, followed by other most common fusions in *NF1*, *CDKN2A*, *RAF1*, and *ETV6* genes [4,5,6].

The efficacy of targeting fusion alterations in malignant melanoma is unknown, as many gene fusions are uncharacterized either biologically or by clinical response to targeted therapy. In vitro, functional studies showed sensitivity of *RAF1* and *BRAF* fusions to MEK inhibitors [7]. A patient harboring an *ANO10-RAF1* fusion displayed sensitivity to trametinib [8], and another melanoma harboring a *GOLGA4-RAF1* fusion was associated with increased ERK activation and significant response to MEK inhibitor treatment [7]. Clinical response to trametinib was reported in two patients with *BRAF* fusion-positive metastatic melanoma [9]. Patient-derived melanoma cell lines expressing an *AGK-BRAF* fusion also showed sensitivity to sorafenib, with a durable response observed in the corresponding fusion-positive patient [10].

To further describe the incidence and relevance of fusions in malignant melanoma, we retrospectively reviewed comprehensive molecular profiles from a large real-world cohort of patients with melanoma. In addition to the discovery of previously unreported gene fusions, gene expression profiles were further analyzed to elucidate the functional consequences of gene fusions in this aggressive malignancy. The results may better inform treatment decisions and identify new strategies for targeted therapies and immunotherapy.

## 2. Materials and Methods

Clinical physicians worldwide submitted formalin-fixed paraffin-embedded (FFPE) samples (n = 1255) from patients with melanoma (n = 1243) to a commercial CLIA-certified laboratory for molecular profiling (Caris Life Sciences, Phoenix, AZ) from February 2019 to July 2020. The present study was conducted in accordance with the guidelines of the Declaration of Helsinki, Belmont Report, and US Common Rule. In compliance with policy 45 CFR 46.101(b), this study was conducted using retrospective, de-identified clinical data, and patient consent was not required.

DNA Next-Generation Sequencing (NGS) of 592 cancer-relevant genes was performed on genomic DNA isolated from formalin-fixed paraffin-embedded (FFPE) tumor samples using the NextSeq platform (Illumina, Inc., San Diego, CA, USA). Matched normal tissue or germline DNA was not sequenced. A custom-designed SureSelect XT assay was used to enrich exonic regions of 592 whole-gene targets (Agilent Technologies, Santa Clara, CA, USA). All variants were detected with >99% confidence based on allele frequency and amplicon coverage, with an average sequencing depth of coverage of >500 and an analytic sensitivity threshold established of 5% for variant calling. Prior to molecular testing, tumor enrichment was achieved by harvesting targeted tissue using manual microdissection techniques. Genomic variants were classified by board-certified molecular geneticists according to criteria established by the American College of Medical Genetics and Genomics (ACMG). When assessing mutation frequencies of individual genes, ‘pathogenic’, and ‘likely pathogenic’ were counted as mutations, while ‘benign’, ‘likely benign’ variants, and ‘variants of unknown significance’ were excluded.

RNA Whole Transcriptome Sequencing (WTS) uses a hybrid-capture method to pull down the full transcriptome from FFPE tumor samples using the Agilent SureSelect Human All Exon V7 bait panel (Agilent Technologies, Santa Clara, CA, USA) and the Illumina NovaSeq platform (Illumina, Inc., San Diego, CA, USA). FFPE specimens underwent pathology review to discern the percent tumor content and tumor size; a minimum of 20% tumor content in the area for microdissection was required to enable enrichment and extraction of tumor-specific RNA. A Qiagen RNA FFPE tissue extraction kit was used for extraction, and the RNA quality and quantity were determined using the Agilent TapeStation. Biotinylated RNA baits were hybridized to the synthesized and purified cDNA targets, and the bait-target complexes were amplified in a post-capture PCR reaction. The resultant libraries were quantified and normalized, and the pooled libraries were denatured, diluted, and sequenced. Raw data were demultiplexed using the Illumina DRAGEN FFPE accelerator. FASTQ files were aligned with STAR aligner (Alex Dobin, release 2.7.4a GitHub). A full 22,948-gene dataset of expression data was produced by the Salmon, which provides fast and bias-aware quantification of transcript expression [11]. BAM files from STAR aligner were further processed for RNA variants using a proprietary custom detection pipeline. The reference genome used was GRCh37/hg19, and analytical validation of this test demonstrated ≥97% Positive Percent Agreement (PPA), ≥99% Negative Percent Agreement (NPA), and ≥99% Overall Percent Agreement (OPA) with a validated comparator method. Identified fusion transcripts were further evaluated to determine breakpoint positions and functional domains retained from a fused gene. Fusions were classified as pathogenic by board-certified molecular geneticists according to criteria established by the ACMG. Novel fusions included those not previously reported at the time of the literature review.

Pathway alterations were defined as a ‘pathogenic’ or ‘likely pathogenic’ variant detected in one or more genes associated with each pathway: MAPK (*ARAF*, *BRAF*, *RAF1*, *MAP2K1*, *MAP2K2*, *MAP2K4*, *MAP3K1*, *HRAS*, *KRAS*, *NRAS*, and *NF1*), PI3K/AKT/MTOR (*PIK3CA*, *PIK3R1*, *AKT1*, *MTOR*, *PTEN*, *TSC1*, and *TSC2*), WNT/Beta-catenin (*APC*, *CTNNB1*, and *RNF43*), DNA Damage Response (*ATM*, *ATRX*, *BAP1*, *BARD1*, *BLM*, *BRCA1*, *BRCA2*, *BRIP1*, *CDK12*, *CHEK2*, *ERCC2*, *FANCA*, *FANCC*, *FANCD2*, *FANCE*, *FANCF*, *FANCG*, *FANCL*, *MLH1*, *MRE11*, *MSH2*, *MSH6*, *MUTYH*, *NBN*, *PALB2*, *PMS2*, *POLE*, *PRKDC*, *RAD50*, and *WRN*), and Cell Cycle Regulation (*CDKN2A*, *CDKN1B*, *CCND1*, *CDK4*, *RB1*, and *TP53*; also includes copy number amplifications [≥6 copies] of *CCND1*, *CDK4*, *MDM2*, and *MYC*).

T cell-inflamed scores were defined by an 18-gene signature, with scores calculated as the weighted sum of log2-transformed gene expression values using previously reported coefficients [12]. The Microenvironment Cell Populations (MCP)-counter tool (Becht 2016) was used to assess the relative abundance of immune and stromal cells in the tumor microenvironment. The MAPK pathway activation score was calculated using a 10-gene set (*SPRY2*, *SPRY4*, *ETV4*, *ETV5*, *DUSP4*, *DUSP6*, *CCND1*, *EPHA2*, and *EPHA4*) previously reported to correlate with clinical outcomes in melanoma and other cancers [13].

Immunohistochemistry (IHC) was performed on full formalin-fixed paraffin-embedded (FFPE) sections of glass slides. Slides were stained using the Agilent DAKO Link 48 (Santa Clara, CA, USA) automated platform and staining techniques, per the manufacturer’s instructions, and were optimized and validated per CLIA/CAP and ISO requirements. Staining was scored for intensity (0 = no staining; 1+ = weak staining; 2+ = moderate staining; 3+ = strong staining) and staining percentage (0–100%). PD-L1 antibody (SP142 or 28-8 clones) staining results were categorized as positive (≥1+ and ≥1% tumor cells) or negative (0 or 0%). For nine fusion-positive patient samples with available tissue, MAPK pathway activation was evaluated using antibodies to phospho-ERK1/2 (Thr202/Tyr204; Cell Signaling Technologies [CST] #4370) and total-ERK1/2 protein (CST #9102), with the proportion of phosphorylated protein determined by the min-max normalized ratio phospho:total-ERK1/2 H-scores (stain intensity * percentage of cells stained), in comparison to *BRAF*-V600E and MAPK pathway-WT control samples (WT indicates no MAPK pathway alterations detected).

Tumor Mutational Burden (TMB): TMB was measured by counting all non-synonymous missense, nonsense, in-frame insertion/deletion, and frameshift mutations found per tumor that had not been previously described as germline alterations in dbSNP151, Genome Aggregation Database (gnomAD) databases, or benign variants identified by Caris’s geneticists. A cutoff point of ≥10 mutations per megabase (mt/MB) was used based on the KEYNOTE-158 pembrolizumab trial [14]. Caris Life Sciences is a participant in the Friends of Cancer Research TMB Harmonization Project [15].

All statistical analyses were performed with JMP V13.2.1 (SAS Institute, Cary, NC, USA), or R Version 3.6.1 (https://www.R-project.org, accessed on 3 February 2020). Continuous data were assessed using a Mann–Whitney U test, and categorical data were evaluated using Chi-square or Fisher’s exact test, where appropriate. *p*-values were adjusted for multiple hypothesis testing using the Benjamini–Hochberg procedure.

## 3. Results

### 3.1. Patient Cohort Characteristics

The sequencing results from a total of 1255 molecularly profiled tumor samples from 1243 patients with melanoma were retrospectively analyzed. The study was composed of 61.9% male patients (n = 777) and 63.1% metastatic biopsies (n = 780), with an overall median age of 67 years (range 3–90+) (Table 1). Pathogenic/likely pathogenic (P/LP) in-frame fusion events were present in 33 (2.6%) cases, 14 of which were novel fusions not previously reported, while fusion transcripts with unknown pathogenicity were detected in 669 (53.3%) melanoma samples. A total of 2404 unclassified fusion isoforms were detected, including 10 recurrent (n ≥ 3) fusions (Appendix A). There was an overall average of 1.6 fusions per tumor, and an average of 2.9 fusions among all fusion-positive samples, regardless of known pathogenicity. Table 1 includes a comparison between fusion-positive and fusion-negative cohort characteristics.

### 3.2. Key Pathway Alterations and Therapy-Associated Biomarkers in Fusion-Positive Tumors

We next evaluated tumor molecular profiles to determine if common biomarkers and pathway alterations (defined as a mutation in one or more pathway-associated genes noted in the methods) were associated with fusions in malignant melanoma. MAPK pathway alterations were significantly less frequent in tumors harboring fusions (Figure 1), with non-*BRAF* p.V600X mutations comprising the majority of MAPK pathway alterations (7/8) in tumors with P/LP fusions. Tumors with P/LP fusions had lower rates of PD-L1+ expression compared to fusion-negative tumors (26.1 vs. 45.9%, *p* = 0.0619), which was consistent with a reduced rate of high tumor mutational burden (TMB-H, ≥10 mutations/Mb) (33.3 vs. 53.3%, *p* = 0.0258). Other pathway alteration rates, including that of the DNA damage response pathway, were similar in tumors with and without detected fusions.

### 3.3. Functional Domains in Oncogenic Fusions and Co-Alterations

To better understand the functional consequences of presumed oncogenic gene fusion events, we first determined which functional domains were retained from each fusion partner. Of the 33 oncogenic fusion-positive samples identified, oncogene fusion partners included *BRAF* (n = 21), *RAF1* (n = 4), *PRKCA* (n = 4), *TERT* (n = 2), *AXL* (n = 1), and *FGFR3* (n = 1), 14 of which have not been previously reported (Figure 2). A variety of genes with different functional domains were fused with *BRAF*, while a recurrent fusion with *AGK* (n = 4) had no known functional domains contributed by the short *AGK* sequence fused with *BRAF* (Figure 2A,C). No recurrent fusion partners were observed for *RAS1* or other oncogenes, most of which were novel fusions. With the exception of *TERT* fusions, all presumed oncogenic fusions retained the tyrosine kinase domain from the oncogene partner.

TMB-H and PD-L1+ expression were each observed in six *BRAF* fusion-positive tumors, yet concurrently in only two tumors (Figure 2B,D). TMB-H was also observed in *RAF1* fusion-positive melanomas (n = 3), with pathogenic mutations identified in two or more genes per tumor. *NRAS* (n = 3) and *NF1* co-mutations (n = 2) were identified in *PRKCA* fusion-positive tumors, and one tumor harbored a *PRKCA* fusion with *NF1*. Tumors with *TERT* fusions had co-mutations in *BRAF* (n = 1) or *NRAS* (n = 1). While no co-alterations were identified in an *FGFR3* fusion-positive tumor, an *AXL* fusion-positive tumor was *TP53*-mutated and PD-L1+. Notably, *BRAF* and *RAF1* fusion-positive tumors lacked concurrent MAPK pathway alterations.

### 3.4. BRAF and RAF1 Fusion-Positive Tumors Exhibit MAPK Pathway Activation in the Absence of Other Driver Alterations

Although inhibitors of the MAPK pathway remain a common therapeutic strategy in malignant melanoma, many trials have focused on patients harboring *BRAF*-V600E mutations, without similar enrollment opportunities for patients with *BRAF* fusion-positive tumors. To determine if MAPK signaling was elevated in tumors with fusions, we assessed a transcriptional signature of MAPK pathway activity that has been demonstrated to predict sensitivity to cobimetinib in cell lines representing multiple cancer types and sensitivity to vemurafenib in melanoma patients (12). Despite the absence of concurrent known driver mutations of MAPK signaling, the median MAPK pathway activation score of tumors with *BRAF* and *RAF1* fusions were significantly higher than those without MAPK pathway alterations (MAPK pathway-WT), consistent with tumors harboring *BRAF*-V600X and *NRAS* mutations (Figure 3). Significantly elevated MAPK signaling was also observed in *TERT* and *PRKCA* fusion-positive tumors, which harbored concurrent *BRAF*, *NRAS*, or *NF1* mutations.

In support of MAPK pathway activation, IHC analysis for select samples with available tissue revealed increased phosphorylation of ERK1/2 in *BRAF* and *RAF1* fusion-positive tumors, as well as those with *PRKCA* fusions (Appendix A). However, in *AXL* and *FGFR3* fusion-positive tumors, no concurrent MAPK pathway alterations were identified, and MAPK pathway activation scores were similar to MAPK pathway-WT tumors (Figure 3). Together, these results suggest *BRAF* and *RAF1* fusions are drivers of MAPK signaling in melanoma. Functional studies are needed to assess the benefit of MAPK pathway inhibitor therapies.

We further assessed MAPK pathway activation for samples harboring the recurrent unclassified fusions with unknown oncogenicity (Appendix A). *MTAP:CDKN2B-AS1*, *RIPK1:SERPINB9*, and *LYST:NID1* fusion-positive samples had significantly increased MAPK pathway activation scores compared to MAPK pathway-WT samples. However, 82.9% (29/35) of these samples harbored concurrent MAPK pathway alterations, suggesting the fusions are unlikely to be drivers MAPK signaling in melanoma.

### 3.5. Melanomas with Oncogenic Fusions Events Display Variable Signaling Pathway Activation and Tumor Immune Cell Infiltrates

We next performed gene expression profiling of the tumor microenvironment (TME) to determine the potential sensitivity of fusion-positive melanoma to immunotherapies. Cell type-specific gene sets were used to estimate the relative abundance of cell populations infiltrating the TME. Increased T cell-inflamed scores were associated with PD-L1 expression by IHC in fusion-positive and fusion-negative tumors (Figure 4A). T cell-inflamed scores in tumors harboring pathogenic/likely pathogenic fusions were not significantly different from MAPK pathway-WT tumors (Figure 4B). Moreover, similar correlations between T cell-inflamed scores and immune/stromal cell population abundances were observed for fusion-positive and fusion-negative samples. T cell-inflamed scores correlated with immune/stromal population abundances in the tumor microenvironment (TME) among fusion-positive and fusion-negative tumors, indicating the scores reflect similar changes in TME. Together, this suggests that, while T cell-inflamed scores reflect similar differences in TME composition among fusion-positive and fusion-negative samples, pathogenic/likely pathogenic fusions are not predictive of sensitivity to immunotherapies.

## 4. Discussion

Herein, we report our findings from a large, real-world cohort of molecularly profiled melanoma patient samples. Pathogenic/likely pathogenic oncogenic fusions were uncommon, occurring in only 2.6% of melanomas. Among tumors with oncogenic fusions, those harboring *BRAF* and *RAF1* fusions exhibited increased MAPK pathway activation despite the absence of a concurrent driver mutation, suggesting these patients could potentially benefit from MAPK inhibitor therapies after further functional studies are performed. Consistent with these findings, activating *RAF1* fusions were recently identified in *BRAF*/*NRAS*/*NF1*-wild type melanomas [16], and fusions involving *BRAF*, *RAF1*, and *ALK* genes have been reported from a cohort of driver-negative Chinese patients with melanoma [17]. Conversely, oncogenic fusions were associated with high and low T cell-inflamed scores, often consistent with PD-L1 IHC status (Figure 4A) indicating that fusions themselves may not reliably predict response to immunotherapy for melanoma. Our findings highlight the need for more functional studies to assess the impact of these fusions, which can lead to improved precision oncology approaches to treatment planning and inclusion of fusion-positive patients in the design of future clinical trials.

Oncogenic fusions have been reported across cancer types in both pediatric and adult patients. For example, *RAF1* and *BRAF* fusions occur in pediatric hematological malignancies, brain tumors, sarcomas, melanomas, and other cancers [18], while fusions in oncogenes such as *RAF*, *RET*, *ALK*, *NTRK*, and *FGFR* have been reported in adult epithelial tumors, with some tumor-specific gene fusions defined as therapeutic, diagnostic, or prognostic biomarkers [19]. *ALK-EML4* fusions have been reported in 2–7% of patients with advanced non-small-cell lung cancer (NSCLC) [20], and clinical trials are ongoing to determine if ALK inhibitors are currently approved for use in NSCLC can effectively target these fusions and overcome resistance. In addition to NSCLC and certain types of lymphomas, *ALK-EML4* fusions have been reported in inflammatory myofibroblastic tumors, papillary thyroid cancer, breast cancer, and colorectal cancer [21,22,23]. Studies using patient-derived colorectal cancer organoids to compare *BRAF* fusions with various fusion partners (*TRIM24*, *AGAP3*, and *DLG1*) have shown that the 5′ partner plays a role in signaling and localization that affects signaling pathways and gene expression [24]. Clinical data revealed that tumors with *SEPT3-BRAF* fusions had a more aggressive nature when compared to *ARMC10-BRAF* and *AGK-BRAF* fusions. While oncogenic fusions typically have an intact kinase domain that is comparable to *BRAF* wild type, thus limiting the efficacy of selective BRAF inhibitors, targeting activated pathways downstream to the activated kinase fusion with MEK inhibitors may be effective [25]. Thus, detection of potentially oncogenic fusions and a deeper understanding of their biological function in melanoma are vital for personalized precision medicine. Further analysis of fusion-positive tumors for gene expression profiles suggestive of MAPK pathway activation is expected to more reliably predict for responders of targeted therapy, and future studies should continue to explore the effect of gene fusions and gene expression profiles as a composite biomarker in melanoma.

In addition to the retrospective nature of our study, limitations include a lack of certain demographic information (e.g., race/ethnicity), detailed clinical history and staging, and treatment outcome in response to targeted therapies. While the relative differences among tumor gene expression profiles provide important insights into the underlying biology of fusion-positive melanoma, interpretation of results is limited by the lack of matched normal tissue sequencing data. Although clinical trials are underway to investigate the targetability of fusions in melanoma, future clinical trials should consider the inclusion of tumor profiling by WTS to identify gene fusion events that may not be detected by targeted sequencing panels.

## 5. Conclusions

Our results show that, while oncogenic gene fusions are relatively rare events in melanoma, they are associated with gene expression profiles indicating MAPK pathway activation, suggesting they could cautiously be studied for targeted therapies with currently available MAPK pathway inhibitors. Despite reports of clinical response to targeted therapy in melanoma patients harboring fusions, the efficacy of targeting fusions remains unclear, and additional analyses are needed to functionally characterize the oncogenic effect of these fusions. Future studies should continue to investigate the role of gene fusions in melanoma that may benefit the design of biomarker-driven clinical trials.

## Figures and Tables

**Figure 1 cancers-14-01505-f001:**
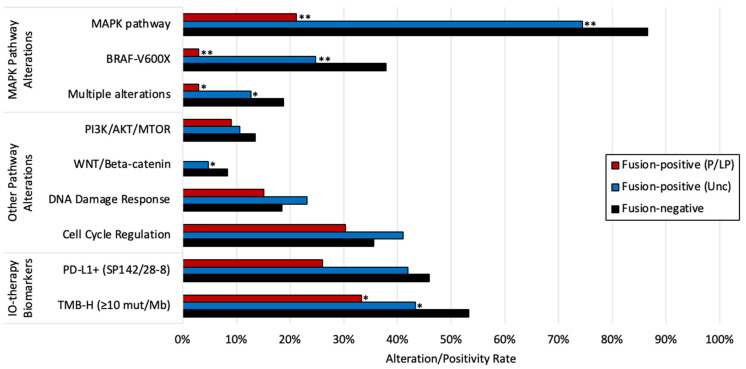
Prevalence of key pathway alterations and therapy-associated biomarkers in fusion-positive and fusion-negative tumors. * *p* < 0.05, ** Q < 0.05 (Benjamini–Hochberg). *p*-values reflect the comparison with the fusion-negative cohort.

**Figure 2 cancers-14-01505-f002:**
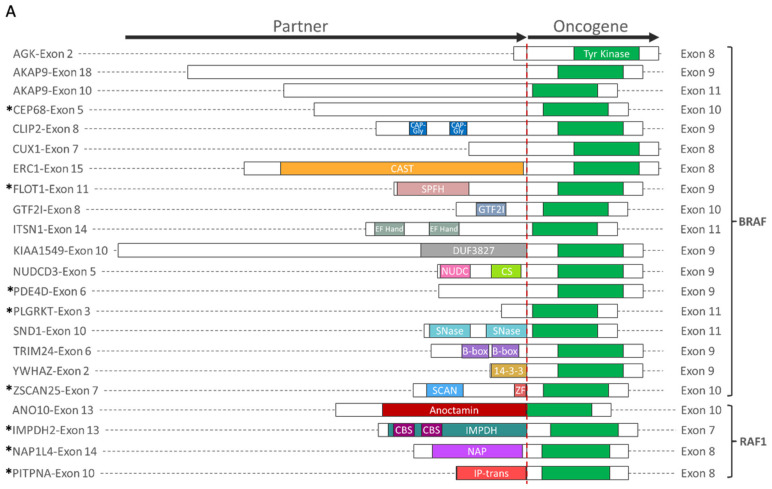
Oncogenic fusion schematics and co-alterations. (**A**,**C**) Schematics of gene fusions with known functional domains annotated. (**B**,**D**) Co-alterations identified in fusion-positive samples. RAS family fusions shown in (**A**,**B**), and fusions involving *PRKCA*, *TERT*, *AXL*, and *FGFR3* shown in (**C**,**D**). * Novel fusion.

**Figure 3 cancers-14-01505-f003:**
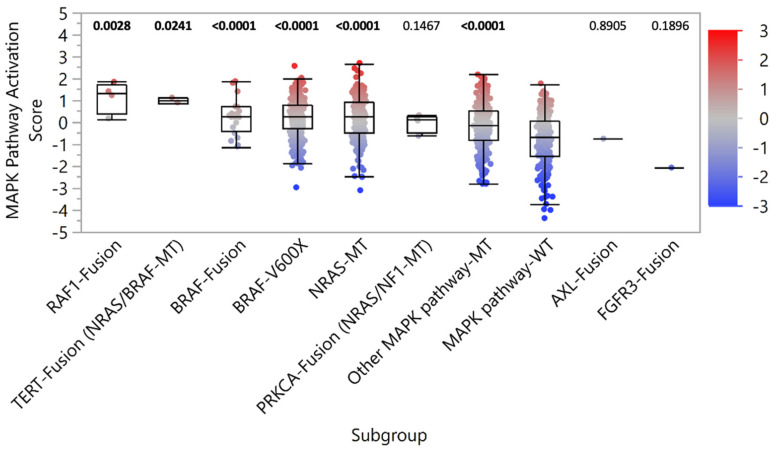
MAPK pathway activation. MAPK pathway activation scores were determined by the relative expression of key gene transcripts. Patient samples were stratified into subgroups based on the detection of pathogenic/likely pathogenic fusion or MAPK pathway alteration (co-alterations noted in parentheses). *p*-values (Mann-Whitney U) noted above each subgroup reflect the comparison with the MAPK pathway-WT control subgroup.

**Figure 4 cancers-14-01505-f004:**
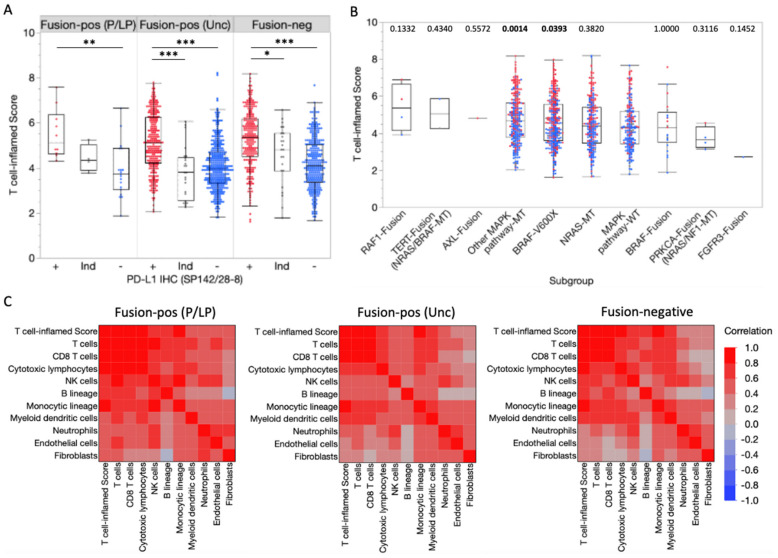
Transcriptional profiling of the tumor microenvironment. (**A**) T cell-inflamed scores in fusion-positive and fusion-negative tumors according to PD-L1 IHC (SP142/28-8) status (Red = positive [+], Blue = negative [–], gray = Indeterminate [Ind]). * *p* < 0.05, ** *p* < 0.01, *** *p* < 0.0001. (**B**) T cell-inflamed scores in subgroups based on detected fusions or MAPK pathway alterations (co-alterations noted in parentheses). *p*-values (Mann-Whitney U) noted above each subgroup reflect the comparison with the MAPK pathway-WT control subgroup. (**C**) Spearman correlation coefficients for T cell-inflamed scores and immune/stromal cell population abundances.

**Table 1 cancers-14-01505-t001:** Patient cohort characteristics. Fusion-positive samples include those with one or more unique fusion transcripts detected, which were further stratified based classification of fusion transcripts as pathogenic/likely pathogenic (P/LP) or unclassified (Unc). *p*-values reflect the comparison with the fusion-negative cohort.

Characteristic	All Cases	Fusion-Positive (P/LP)	Fusion-Positive (Unc)	Fusion-Negative
Total, N samples (% of total)	1255 (100%)	33 (2.6%)	669 (53.3%)	553 (44.1%)
Median Age, years (SD)	67 (13.5)	60 (16.1)	68 (12.9)	66 (14.0)
Age Range, years	3–90	23–84	23–90	3–90
*p*-value (Mann–Whitney U)	------	0.0995	0.0071	------
Female/Male, N cases	478/777	16/17	277/392	185/368
(% Female/% Male)	(38.1%/61.9%)	(48.5%/51.5%)	(41.4%/58.6%)	(33.5%/66.5%)
*p*-value (Chi-square)	------	0.0772	0.0043	------
Metastatic/Primary, N cases	780/456	21/12	411/246	348/198
(% Metastatic/% Primary)	(63.1%/36.9%)	(63.6%/36.4%)	(62.6%/37.4%)	(63.7%/36.3%)
N unclear	19	0	12	7
*p*-value (Chi-square)	------	0.9908	0.673	------

## Data Availability

No publicly archived datasets were used for this study.

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
