# Peer review of "Transcriptional Profiling of Malignant Melanoma Reveals Novel and Potentially Targetable Gene Fusions"

_cancers, 2022, doi:10.3390/cancers14061505_

Round 1

Reviewer 1 Report

The authors performed retrospective review of patient samples that underwent next-generation sequencing of the exons of cancer-relevant genes and whole transcriptome sequencing for the detection of gene fusion events and gene expression profiling. Although the proportion of tumors with pathogenic/likely pathogenic oncogenic fusions is small, many of them are associated with activation of the MAPK pathway, suggesting that they may be potential therapeutic targets in the future. This study seems to be worth reporting, but I think several points need to be explained more clearly.

Introduction

Page 2, line 49-50

The statements that “many melanomas do not harbor targetable genomic alterations” is vague and left open to reader interpretation. I think that not a small number of patients harbored BRAF V600E/K mutation, although the percentage varies by race or countries.

Result

Page 4, line179-180, Table 1

This statement means that multiple tumor samples were obtained from the same patient, but patient information such as age and sex were counted in duplicate. Were these tumor samples obtained from a patient with multiple primary tumors? If so, was there a clear distinction between them and skin metastases? Or did you collect different lymph node or organ metastases from the same patient? Although it is assumed that primary and metastatic lesions in the same patient may have some discrepancies in genomic alterations, it does not seem appropriate to analyze them as completely different samples. In addition, melanoma is known to have several different clinical subtypes, and each clinical subtype tends to have a different mutational profile. Is it possible to add information about clinical subtypes in this study?

Table 1 and Figure 1

Is the fusion-positive (Unc) group worthy of comparison to the others in this study? Since this study focuses on potential therapeutic gene fusions, it would be more understandable to compare tumors with and without pathogenic gene fusions.

Figure 4

I think it is difficult to understand the characteristics of the tumor with oncogenic fusions from this figure. Can you explain how the control tumors were randomly selected? Also, can you explain more clearly the definition of subgroup classification? Depending on how you show the data, it may look like there are a lot of "cold tumors" in the tumors with fusions. In addition, I ask this simply out of curiosity, was it possible to assess the tumor microenvironment in all samples? For lymph node metastasis, the surrounding tissue is removed and only the lymph nodes are submitted for pathology in our country. If it is not a problem, it is not necessary to include this information in the text.

Reviewer 2 Report

Authors should describe the influence of FFPE sample in quality of RNA analysis. Are all samples that authors extracted had good quality?

Authors described “While tumors with P/LP fusions had similar rates 200 of PD-L1+ expression as fusion-negative tumors (26.1 vs 45.9%, P=0.0619)”. But, I do not feel similar although they are not significantly different.

Why does PD-L1 fusion positive tumors showed lower TMB?

Authors should add discussion about Figure 4. What is the reason why most BRAF and PRKCA fusion-positive tumors were associated with a cold TME and RAF1 fusion-positive tumors exhibited intermediate and hot?

Round 2

Reviewer 2 Report

I have no more comments.